# Within-Modality Synthesis and Novel Radiomic Evaluation of Brain MRI Scans

**DOI:** 10.3390/cancers15143565

**Published:** 2023-07-10

**Authors:** Seyed Masoud Rezaeijo, Nahid Chegeni, Fariborz Baghaei Naeini, Dimitrios Makris, Spyridon Bakas

**Affiliations:** 1Department of Medical Physics, Faculty of Medicine, Ahvaz Jundishapur University of Medical Sciences, Ahvaz, Iran; masoudrezayi1398@gmail.com (S.M.R.);; 2Faculty of Engineering, Computing and the Environment, Kingston University, Penrhyn Road Campus, Kingston upon Thames, London KT1 2EE, UK; f.baghaeinaeini@kingston.ac.uk (F.B.N.); d.makris@kingston.ac.uk (D.M.); 3Richards Medical Research Laboratories, Center for Biomedical Image Computing and Analytics (CBICA), University of Pennsylvania, Floor 7, 3700 Hamilton Walk, Philadelphia, PA 19104, USA

**Keywords:** MRI synthesis, T2W, FLAIR, radiomic, CycleGAN, DC^2^Anet

## Abstract

**Simple Summary:**

Brain MRI scans often require different imaging sequences based on tissue types, posing a common challenge. In our research, we propose a method that utilizes Generative Adversarial Networks (GAN) to translate T2-weighted-Fluid-attenuated-Inversion-Recovery (FLAIR) MRI volumes into T2-Weighted (T2W) volumes, and vice versa. To evaluate the effectiveness of our approach, we introduce a novel evaluation schema that incorporates radiomic features. We train two distinct GAN-based architectures, namely Cycle GAN and Dual Cycle-Consistent Adversarial network (DC2Anet), using 510 pair-slices from 102 patients. Our findings indicate that the generative methods can produce results similar to the original sequence without significant changes in radiometric features. This method has the potential to assist clinicians in making informed decisions based on generated images when alternative sequences are unavailable, or time constraints prevent re-scanning MRI patients.

**Abstract:**

One of the most common challenges in brain MRI scans is to perform different MRI sequences depending on the type and properties of tissues. In this paper, we propose a generative method to translate T2-Weighted (T2W) Magnetic Resonance Imaging (MRI) volume from T2-weight-Fluid-attenuated-Inversion-Recovery (FLAIR) and vice versa using Generative Adversarial Networks (GAN). To evaluate the proposed method, we propose a novel evaluation schema for generative and synthetic approaches based on radiomic features. For the evaluation purpose, we consider 510 pair-slices from 102 patients to train two different GAN-based architectures Cycle GAN and Dual Cycle-Consistent Adversarial network (DC^2^Anet). The results indicate that generative methods can produce similar results to the original sequence without significant change in the radiometric feature. Therefore, such a method can assist clinics to make decisions based on the generated image when different sequences are not available or there is not enough time to re-perform the MRI scans.

## 1. Introduction

Medical imaging scans, including Magnetic Resonance Imaging (MRI) and Computed Tomography (CT), are routinely acquired and used clinically to macroscopically assess, diagnose, and monitor patients with brain abnormalities. MRI in particular can depict normal anatomy and apparent pathologies while providing data relating to the anatomical structure, tissue density, and microstructure, as well as tissue vascularization, depending on the acquired sequence [1,2,3]. Structural MRI sequences represent the basic scans acquired across comprehensive centers and community-based healthcare sites, comprising native T1-weighted (T1W), post-contrast T1-weighted (T1Gd), T2-weighted (T2W), and T2-weighted Fluid-Attenuated Inversion Recovery (FLAIR). T1W scans facilitate observation and analysis of the brain anatomy, with the T1Gd scans particularly being able to easily identify the boundaries of an active tumor, while T2-weighted scans (T2w and FLAIR) help in identifying brain abnormalities, both those related to vascular lesions (e.g., stroke) and vasogenic edema [4]. The simultaneous assessment of multiple varying MRI scans (also known as multi-parametric MRI—mpMRI) from the same patient is the standard clinical practice for the evaluation of patients suspected of stroke or diffuse glioma, as it offers the maximal available medical diagnostic information.

Acquisition of mpMRI might not be possible at all times, due to numerous reasons, including but not limited to the patient’s cooperation during a scanning session that could result in motion-degraded scans, thereby hindering further diagnostic usage [5,6,7]. Towards this end, the artificial synthesis of specific MRI scans has been an active area of research [8,9], with the intention of either substituting specific MRI scans corrupted by various artifacts, or generating scans that were not acquired at all. Although such synthetic scans have been successfully used in many applications, and Generative Adversarial Networks (GANs) have significantly improved their realism, the final result may not always look as realistic and/or may contain information that adversely affects downstream quantitative analyses [10,11]. Cross-domain synthesis of medical images has drawn significant interest in the medical imaging community and describes the artificial generation of a target-modality scan by learning the relationship between paired given source-modality scans and their associated target modality scans [12,13]. Of note, the data is here described as paired when it arises from the same individual at different points in time. 

In recent years, deep-learning procedures and particularly Convolutional Neural Networks (CNNs) [14] and GANs have rapidly dominated the domain of medical image synthesis [15,16]. GANs use two competing CNNs: one that generates new images and another that discriminates the generated images as either real or fake. To address the problem of unpaired cross-domain data, which is common in healthcare, the Cycle Generative Adversarial Network (CycleGAN) [17] is typically chosen to obtain high-quality information translatable across images. In CycleGAN, based on the image of a subject b_1_ in the source domain, the purpose is to estimate the relevant image of the same subject b_2_ in the target domain. In theory, the CycleGAN model entails two mapping functions, i.e., G_1_: X → Y and G_2_: Y → X, and associated adversarial discriminators D_Y_ and D_X_. D_Y_ encourages G_1_ to translate X into outputs indistinguishable from domain Y and contrariwise for D_X_ and G_2_. Nie et al. [18] trained a fully convolutional network (FCN) to generate CT scans from corresponding MRI scans. They specifically used the adversarial training method to train their FCN. Welander et al. [19] evaluated two models, i.e., CycleGAN and UNIT [20], for image-to-image translation of T1- and T2W MRI slices by comparing synthetic MRI scans to real ones. They used paired T1W and T2W images from 1113 axial images (only slice 120). The scans were registered to a standard anatomical template, so they were in the same coordinate space and of the same size. Two models were compared using quantitative metrics, including mean absolute error (MAE), mutual information (MI), and peak signal-to-noise ratio (PSNR). It was shown that the executed GAN models can synthesize visually realistic MRI slices. Dar et al. [21] proposed a method for multi-contrast MRI synthesis based on Conditional GANs (CGANs). They demonstrated how CGANs could generate a T2W scan from a T1W. Theis et al. [22] found that GANs can generate more realistic training data to improve the classification performance of machine learning methods. Also, they showed that models creating more visually realistic synthetic images do not certainly have better quantitative error measurements when compared to real images. Despite the mounting promise of GANs for healthcare, both optimal model selection and quantitative evaluation remain challenging tasks, and solutions produced so far are use-specific and not generalizable. Specifically, for their quantitative performance evaluation, several metrics (such as the MAE, Mean Squared Error (MSE), and PSNR) have been proposed in the literature [23], albeit no consensus has been reached on an optimal evaluation metric for a particular domain. 

Radiomics describe a novel and rapidly advancing area in medical imaging. In contrast to the traditional clinical assessment of considering medical images as pictures intended only for visual interpretation, radiomics represent visual and sub-visual quantitative measurements (also called “features”) extracted from acquired radiology scans [24,25], following specific mathematical formulations, and resulting in measurements that are not even perceivable by the naked eye, i.e., sub-visual [26,27,28,29,30,31,32,33,34]. These features are widely used in both clinical and pre-clinical research studies attempting to identify associations between radiologic scans and clinical outcomes, or even molecular characteristics [35,36,37,38]. The hypothesis is that quantitative computational interrogation of medical scans can provide more and better information than the physician’s visual assessment. This is further exacerbated in observations related to the texture analyses of different imaging modalities. Various open-source tools have been developed to facilitate the harmonized extraction of high throughput radiomic features [39,40,41], contributing to the increasing evidence of their value. The primary purpose of these tools has been to expedite robust quantitative image analyses based on radiomics and standardize both feature definitions and computation strategies, thereby guaranteeing the reproducibility and reliability of radiomic features. In this study, considering the importance of T2-weighted scans (T2W and FLAIR), we focus on generating FLAIR from T2W MRI scans, and vice versa, based on the CycleGAN [17] and dual cycle-consistent adversarial network (DC^2^Anet) [42] architectures. We further consider radiomics as a novel way to quantify the dissimilarity between the distribution of the actual/real and synthesized scans. We think that radiomics can represent the first potential solution for the quantitative performance evaluation of GANs in the domain of radiology. For comparison, we also compare with traditional metrics, including MSE, MAE, and PSNR [43]. 

## 2. Methods

### 2.1. Dataset and Registration

The data utilized in our study were obtained from the public data collection ‘ACRIN-DSC-MR-Brain (ACRIN 6677/RTOG 0625)’ [44] at The Cancer Imaging Archive (TCIA) [45]. These data describe brain MRI scans from a multicenter phase-II trial of bevacizumab with temozolomide in recurrent glioblastoma (GBM) patients. We extracted 510 T2W/FLAIR paired slices from 102 patients, which were further divided into 410 pairs from 81 patients for training and 100 pairs from 21 patients to evaluate synthesis results (test set). Due to the limitation of the dataset, the validation set is not considered to train the networks. Each pair include the axial paired T2W/FLAIR slices for the same patient and at the same axial depth. Of note, the testing data are held out of the training process at all times. Our networks take 2D axial-plane slices of the volumes as inputs. In the pre-processing step, T2W 2D scans were first rigidly registered to FLAIR scans using the ITK-SNAP software (version 3.8.0) [46], considering 6 degrees of freedom (i.e., 3 translations and 3 rotations).

### 2.2. CycleGAN

A GAN network uses an image generator (G) to synthesize images of a target domain and a discriminator (D) to distinguish between real and synthesized images. A suitable analogy for visual data considers one network as an art forger and the other as an art specialist. The forger (generator G) creates forgeries. The specialist, (discriminator D) receives both forgeries and real images and attempts to tell them apart (Figure 1). Both networks are in competition with each other and are trained simultaneously. 

CycleGAN is a framework that allows the unpaired data to generate an image from one domain to another. Therefore, CycleGAN reduces the problem caused by the lack of paired data. A diagram of the CycleGAN model used in this study is presented in Figure 2. Let us assume n^A^ images x^A^ ∈ XA (e.g., T2W) and n^B^ images x^B^ ∈ XB (e.g., FLAIR). The CycleGAN for T2W and FLAIR images include two mappings G_T2W_: T2W → FLAIR and G_FLAIR_: FLAIR → T2W. Therefore, the proposed CycleGAN operates with two generators (G_T2W_, G_FLAIR_) and two discriminators (D_T2W_, D_FLAIR_). Given a T2W image, G_T2W_ learns to generate the respective FLAIR image of the same anatomy that is indistinguishable from real FLAIR images, whereas D_T2W_ learns to discriminate between synthetic and real FLAIR images. The architecture of the CycleGAN generator is adapted from [17] with 9 residual blocks after early convolutional layers. Similarly, given a FLAIR image, G_FLAIR_ learns to generate the respective T2W image of the same anatomy that is indistinguishable from real T2W images, whereas D_FLAIR_ learns to discriminate between synthetic and real T2W images. We apply adversarial losses to both mapping functions for matching the distribution of generated images to the data distribution in the target domain. For the mapping function G_T2W_: T2W → FLAIR and its discriminator D_T2W_, the objective is expressed as follows:Լ_GAN_ (G_T2W_, D_T2W_, T2W, FLAIR)(1)

Similarly, the adversarial loss is presented for the mapping function G_FLAIR_: FLAIR → T2W and its discriminator D_FLAIR_ as follows:Լ_GAN_ (G_FLAIR_, D_FLAIR_, FLAIR, T2W)(2)

Also, the cycle consistency loss, or Լ_cyc_ (Forward cycle-consistency and Backward cycle-consistency), is used to keep the cycle consistency between the two sets of networks as follows:Լ_cyc_ (G_T2W,_ G_FLAIR_) = [║G_FLAIR_ (G_T2W_(T2W)) − T2W║_1_] + [║G_T2W_ (G_FLAIR_ (FLAIR)) − FLAIR ║_1_](3)

The loss of the whole CycleGAN network is:Լ_CycleGAN_ = Լ_GAN_ (G_T2W_, D_T2W_, T2W, FLAIR) + Լ_GAN_ (G_FLAIR_, D_FLAIR_, FLAIR, T2W) + Լ_cyc_ (G_T2W_, G_FLAIR_)(4)

Figure 3 illustrates the architecture of the generator and discriminator networks within the proposed CycleGAN model, depicting the size and type of each layer.

### 2.3. DC^2^Anet

The DC^2^Anet model, introduced by Jin et al. [42], follows a semi-supervised learning approach that alternates between optimizing supervised and unsupervised learning in order to seek a global minimum for the optimal network. The forward and backward mappings are used to generate the T2W image from a FLAIR image, and vice versa. In the forward cycle-consistent adversarial network with aligned learning, the G_FLAIR_ network generates a synthetic T2W image from a FLAIR image, and this T2W image is then used by the G_T2W_ network to generate the original FLAIR image in order to learn the domain structures. The input to the discriminators Dis_T2W_ is either a sample T2W image from the real T2W data or a synthetic T2W image. In the backward cycle-consistent adversarial network with aligned learning, the G_T2W_ network generates a synthetic FLAIR image from a T2W image, and this FLAIR image is then used by the G_FLAIR_ network to generate the original T2W image in order to learn the domain structures. The discriminator’s Dis_T2W_ and Dis_FLAIR_ are expressed as follows:(5)$$\begin{array}{c} L_{\text{sup-adver}\;}\left({Gen}_{T2W},Dis_{T2W},Gen_{FLAIR},Dis_{FLAIR}\right)=E_{FLAIR,T2W∼p_{\mathrm{data}\;}\left(FLAIR,T2W\right)}\left[\log\left(Dis_{T2W}\left(FLAIR,T2W\right)\right)\right]\\ \;\;\;\;\;\;\;\;\;\;+E_{FLAIR∼p_{\mathrm{data}\;}\left(FLAIR\right)}\left[\log\left(1-Dis_{T2W}\left(FLAIR,{\mathrm{Gen}}_{T2W}\left(FLAIR\right)\right)\right)\right]\\ +E_{T2W,FLAIR∼p_{\mathrm{data}\;}\left(T2W,FLAIR\right)}\left[\log\left(Dis_{FLAIR}\left(T2W,FLAIR\right)\right)\right]\\ \;\;\;\;\;\;\;\;\;\;\;+E_{T2W∼p_{\mathrm{data}\;}\left(T2W\right)}\left[\log\left(1-Dis_{FLAIR}\left(T2W,{\mathrm{Gen}}_{\mathrm{FLAIR}}\left(DWI_{T2W}\right)\right)\right)\right]\; \end{array}$$

In the DC2Anet model, in addition to adversarial and dual cycle consistency used in the CycleGAN model, to achieve accurate and perceptual outputs, four loss functions were measured as follows: voxel-wise, gradient difference, perceptual, and structural similarity losses. We name the loss function of the supervised training of our model as L_sup. Then the value of this function is defined as a weighted sum of L_sup-adversarial_, L_sup-cycle-consistency_, and these four losses (L_voxel-wise_, L_gradient_, L_perceptual_, L_structural_). The weights used to calculate the L_sup_ are hyperparameters of the model, and we set all of them to be one in our experiment. Hence, these four terms are combined, and the relation between them is as follows: L_sup_ = L_sup-adversarial_ + L_sup-cycle-consistency_ + L_voxel-wise_ + L_gradient_ + L_perceptual_ + L_structural_. A diagram outlining the forward and backward adversarial losses of the DC^2^Anet model is shown in Figure 4. Of note, the generator network of the DC^2^Anet model is the same as the CycleGAN model (Figure 3a), but the discriminator network is different. The architecture of this discriminator is shown in Figure 5.

### 2.4. Implementation

To generate a T2W MRI from a FLAIR, and vice versa, the T2W and FLAIR values are converted to [0, 1] tensor. The resolutions of the FLAIR and T2W images in our dataset are 256 × 256 and 512 × 512 respectively. Therefore, in the first preprocessing step, FLAIR images are registered to T2W images using rigid registration to ensure that all images have a 256 × 256 resolution. Then, the axial T2W/FLAIR pairs were the input of the network with 256 × 256 pixels. Of note, instead of image patches, whole 2D images are used for training. CycleGAN and DC^2^Anet were trained for 400 epochs. The batch size was set to 2, and both the generator and the discriminator used the Adam optimizer [47]. In the first 200 epochs, the learning rate was fixed at 2·10^−4^. For the rest 200 epochs, the learning rate linearly decayed from 2·10^−4^ to 0. It was observed that the discriminators found success faster than the generators, therefore the values of iterations for the generator and the discriminator were set to three and one, respectively. The loss function plays a crucial role in training the networks and achieving the desired translation between T2-weighted (T2W) and FLAIR images. In both CycleGAN and DC^2^Anet, adversarial losses are employed to match the distribution of generated images with the target domain data distribution. Specifically, for the mapping function G_T2W_: T2W → FLAIR and its discriminator D_T2W_, we use the adversarial loss defined as Լ_GAN_ (G_T2W_, D_T2W_, T2W, and FLAIR). Similarly, for the mapping function G_FLAIR_: FLAIR → T2W and its discriminator D_FLAIR_, the adversarial loss is expressed as Լ_GAN_ (G_FLAIR_, D_FLAIR_, FLAIR, and T2W). Additionally, to maintain cycle consistency between the generator networks, a cycle consistency loss (Լ_cyc_) is utilized, which ensures that the translated images can be successfully converted back to the original domain. The complete loss function for the CycleGAN network is given by Լ_CycleGAN_ = Լ_GAN_ (G_T2W_, D_T2W_, T2W, FLAIR) + Լ_GAN_ (G_FLAIR_, D_FLAIR_, FLAIR, T2W) + Լ_cyc_ (G_T2W_, G_FLAIR_).

Cyc _(_G_T2W,_ G_FLAIR)_ losses were multiplied by a constant lambda (λ) based on the importance of the cycle consistency losses (Equation (4)) concerning the adversarial loss; therefore, we set λ_cyc (_G_T2W,_ G_FLAIR)_ = 10 and λ_GAN_ = 1. All experiments including data preprocessing and analysis were performed on the Google Cloud computing service “Google Colab” (colab.research.google.com) using Python 3.7 and TensorFlow 2.4.1. The parameters of training and hardware configurations are provided in (Table 1).

### 2.5. Evaluation

Several metrics were used to compare the real and synthetic T2W, and FLAIR, images. These metrics including MAE, MSE, and PSNR were used widely in the literature for the same purpose [48,49,50]. These metrics are defined as:
(6)$$\mathrm{MAE}=\frac{1}{N}\sum_{i=1}^{N}\;⎸\mathrm{real MRI}(\mathrm{T2W}/\mathrm{FLAIR})(i)-\mathrm{synthetic MRI}(\mathrm{T2W}/\mathrm{FLAIR})(i)⎸$$
(7)$$\mathrm{MSE}=\frac{1}{N}\sum_{i=1}^{N}{\;(\mathrm{real MRI}(\mathrm{T2W}/\mathrm{FLAIR})(i)-\mathrm{synthetic MRI}\mathrm{T2W}/\mathrm{FLAIR}(i))}^{2}$$
(8)$$\mathrm{PSNR}={10\cdot \log}_{10}(\frac{{MAX}^{2}}{MSE})$$
where N is the total number of voxels inside the input image and i is the index of the aligned pixel. MAX denotes the largest pixel value of ground truth T2W and synthetic T2W images, and vice versa, for FLAIR images.

Considering a use-inspired generalizable evaluation approach, beyond just the essential quantification but also considering radiologic appearance, in this study we introduce a novel approach based on radiomic features to compare the real and synthetic images. After running the models and generating all the synthetic images, we segmented the whole brain volume, both in the real and synthetic images, using the module of ITK-SNAP [51,52] within the Cancer Imaging Phenomics Toolkit (CaPTk) [41,53]. CaPTk is a software platform written in C++ to analyze medical images. The package leverages the value of quantitative imaging analytics along with machine learning to derive phenotypic imaging signatures. The specific ITK-SNAP module is based on the geometric active contour model and defines the contours using energy forces and the geometric flow curve. The counter is a collection of points that undergo the interpolation process. In this study, manual delineation of the whole brain and skull is performed. Following this segmentation, radiomic features compliant with the Image Biomarker Standardization Initiative (IBSI) [54] were extracted from both the real and synthetic T2W, as well as from the real and synthetic FLAIR images using the CaPTk 1.9.0 software [53,55]. Twenty-three features were extracted, comprising 8 Gray Level Co-occurrence Matrix features (GLCM) [56,57], 8 Gray Level Size Zone (GLSZM) [58], and 7 Gray Level Run Length Matrix features (GLRLM) [54] (Table 2).

The choice of specific features in our study, namely the 8 Gray Level Co-occurrence Matrix (GLCM) features 8 Gray Level Size Zone (GLSZM) features, and 7 Gray Level Run Length Matrix (GLRLM) features, was based on their established significance in characterizing textural patterns and capturing distinct image characteristics (Table 2). These features have been widely utilized in radiomics studies and have demonstrated their efficacy in quantifying spatial relationships, size variations, and run lengths within an image. By utilizing this comprehensive set of radiomic features, we aimed to capture a wide range of textural characteristics that could potentially distinguish real and synthetic images. Details of radiomic features are included in Appendix A.

These features have shown promise in previous studies as reliable indicators of image heterogeneity and structural differences. Their selection was based on their ability to quantitatively represent textural properties and provide discriminative information regarding the underlying tissue or lesion composition. We believe that the inclusion of these 23 selected radiomic features, derived from GLCM, GLSZM, and GLRLM matrices, offers a robust and comprehensive approach for evaluating the differentiation between real and synthetic images. Their relevance lies in their proven capability to capture textural patterns and provide meaningful insights into the image composition, thereby contributing to the assessment and discrimination of real and synthetic images in our study.

The exact parameters used for the feature extraction are: Bins = 20, Radius = 1, and Offset = Combined. Available studies use bin numbers varying from 8 to 1000, as suggested by the IBSI [59], or bin widths from 1 to 75. In this study, textural features were computed using the fixed-bin width approach. Radiomics data were then analyzed using the GraphPad Prism 9.5 software (GraphPad, San Diego, CA, USA). The normality of the extracted radiomic features was evaluated based on the D’Agostino test [60]. When data was described by a normal distribution we used a *t*-test, otherwise we used the Mann-Whitney U test was used. For determining whether the radiomic features between the real and synthetic T2W (as well as between the real and synthetic FLAIR images) are statistically significant, confidence intervals (commonly abbreviated as CI) were calculated.

## 3. Results

We evaluated the proposed CycleGAN and DC^2^Anet architectures on T2W and FLAIR brain tumor images. For the quantitative performance evaluation of the T2W and FLAIR synthesis, in line with current literature we considered the three metrics of MAE, MSE, and PSNR (Table 3).

The CycleGAN model shows superior performance when compared with the DC^2^Anet model across all the 3 quantitative metrics. In general, we observe a better performance for FLAIR images. Of note, smaller values of the MAE and MSE indicate better results, as opposed to PSNR where larger values are better. The quantitative superiority of CycleGAN and DC^2^Anet for FLAIR images corresponds to the visual realness and the mapping in Figure 6 and Figure 7 which show axial views of five patients’ synthetic and real images for CycleGAN and DC^2^Anet, for both the T2W and the FLAIR images. Of note, for both of the T2W and FLAIR images in the perceptual study similar to quantitative analyses, CycleGAN shows the best performance. Hence, the CycleGAN model trained using adversarial and dual cycle consistency generates more realistic synthetic MR images. More examples of CycleGAN and DC^2^Anet for FLAIR and T2W translation are provided in Appendix A.

The differences observed between real and synthesis imaging modalities can be attributed to multiple factors. Firstly, synthesis images are computer-generated simulations of original images based on GAN models, while real original images are directly acquired from patients using MRI scanners. Consequently, variations can arise due to the inherent limitations and assumptions of the synthesis process. Furthermore, physiological and technical factors, such as variances in tissue contrast, signal intensity, and image artifacts, can contribute to dissimilarities between synthesis and real images. To further investigate and address these differences, future studies should focus on refining the synthesis algorithms, incorporating more realistic training data, and exploring the impact of various imaging parameters on the synthesis process.

We then performed a secondary quantitative performance evaluation that considers the radiophenotypical properties of the images, by virtue of the extracted radiomic features. Unlike the metrics of MAE, MSE, and PSNR, significance levels and values of radiomic features vary depending on the type of feature and image. Following a comparison of the radiomic features for both the T2W and FLAIR images, our results pointed out that for most radiomic features, there was no significant difference between the real and synthetic T2W, as well as for the real and synthetic FLAIR images. The mean and standard error (SE) of the GLCM features for both T2W and FLAIR images, as well as their statistically significant difference for the CycleGAN and DC^2^Anet models, are shown in Table 4 and Table 5. No significant differences were observed for all extracted GLCM features using the CycleGAN model between real T2W images and synthetic T2W images. However, this was not the case for three features (cluster prominence, contrast, and correlation) extracted from the synthetic images of the DC^2^Anet model (Table 4). Notably, there was a significant difference for FLAIR images using the DC^2^Anet model for two extracted features (cluster prominence and correlation), and for the cluster prominence feature of the CycleGAN model (Table 5).

Similarly, with GLCM features, no significant differences were observed for all extracted GLRLM features using the CycleGAN model between real T2W images and synthetic T2W images (Table 6). However, with feature extraction on the DC^2^Anet synthetic images, there was a significant difference for two extracted features (High Grey Level Run Emphasis, and Long Run Low Grey Level Emphasis) between the real and synthetic T2W, as well as between the real and synthetic FLAIR images (Table 7).

Extracted features based on GLSZM for the two models used are shown in Table 8 and Table 9. By comparing the real and synthetic T2W based on CycleGAN and DC^2^Anet, except for two features (Grey Level Nonuniformity, and Large Zone Low Grey Level Emphasis), no significant differences were observed for other extracted GLSZM features (Table 8). However, for FLAIR images using the DC^2^Anet model as shown in Table 8, for two features (High Grey Level Emphasis, and Large Zone Low Grey Level Emphasis), significant differences were observed.

## 4. Discussion

A within-modality synthesis strategy was presented for Generating FLAIR images from T2W images and vice versa based on CycleGAN and DC^2^Anet networks. Comprehensive evaluations were conducted for two distinct methods where training images were registered within single subjects. It has been shown, via a perceptual study and in terms of quantitative assessments based on MAE, MSE, PSNR metrics, as well as based on a novel radiomic evaluation, that CycleGAN and DC^2^Anet can be used to generate visually realistic MR images. While our synthesis approaches were primarily evaluated for two specific brain MRI sequences, it has the potential to be applied for image-to-image MRI synthesis, as well as for synthesis across imaging modalities (such as MRI, CT, and PET). The proposed CycleGAN technique uses adversarial loss functions and cycle-consistency loss for learning to synthesize from registered images for improved synthesis. In the DC^2^Anet model, in addition to used losses in the CycleGAN model, four-loss functions, including voxel-wise, gradient difference, perceptual, and structural similarity losses, were used. In modern medical imaging modalities, generating realistic medical images that can be utterly similar to their real ones remains a challenging objective. Generated synthetic images can ensure a trustworthy diagnosis. Based on the quantitative evaluation, for all metrics, the CycleGAN model was found accurate and outperformed the DC^2^Anet model.

CycleGAN and DC^2^Anet models learn the mapping directly from the space of T2W images to the corresponding FLAIR images, and vice versa. Moreover, three metrics including MAE, MSE, and PSNR applied to the data from 510 T2W/FLAIR paired slices from 102 patients, were favorably compared with other reported results in the brain region literature. For example, Krauss et al. [61] compared the assessment results of synthetic and conventional MRIs for patients with Multiple Sclerosis (MS). The images were prospectively acquired for 52 patients with the diagnosed MS. In addition to quantitative evaluations and using the GAN-based approach, the CycleGAN model obtained better results than the study of Krauss et al. Han et al. [62] proposed a GAN-based approach to generate synthetic multi-sequence brain MRI using Deep Convolutional GAN (DCGAN) and Wasserstein GAN (WGAN). Their model was validated by an expert physician who employed Visual Turing test. In agreement with our study, their results revealed that GANs could generate realistic multi-sequence brain MRI images. Nevertheless, this study was different from other research attempts in terms of the employed quantitative evaluation and the proposed models, CycleGAN and DC^2^Anet. In addition to the MAE, MSE, and PSNR metrics, we also conducted a novel evaluation based on radiomic features, to compare the real and synthetic MRI cases. Li et al. [63] proposed a procedure to synthesize brain MRI from CT images. They reported an MAE value of 87.73 and an MSE value of 1.392 × 10^4^ for real MRI and synthesized MRI. Although their results differed from our findings, there was a common understanding that the application of the CycleGAN model was subject to much error. 

For evaluation, generated images and comparing them with real images, quantitative evaluations can be used. It is clear that even if the model achieves a relatively satisfactory score in quantitative measurements including MAE, MSE, and PSNR metrics, it does not necessarily generate visually realistic images. Although visually CycleGAN produced realistic images there is not much difference between the two models CycleGAN and DC^2^Anet in quantitative measurements including MAE, MSE, and PSNR metrics, as shown in Table 3. It can be implied that the process of determining whether or not an image is visually realistic cannot be done based on the mentioned metrics. However, the current study employed radiomic features as a new evaluation approach to compare the real MRI and their synthetic counterparts. Our results (Table 4, Table 5, Table 6, Table 7, Table 8 and Table 9) revealed that for the vast majority of radiomic features regarding the two T2W and FLAIR images, no significant difference was observed between real images and images synthesized using CycleGAN. On the other side, some radiomic features indicate significant differences for images synthesized by DC^2^Anet, indicating that the set of radiomic features is more successful in assessing the realism of the generated images than traditional metrics such as MAE, MSE, and PSNR. Therefore, according to the metrics used in this study, it can be concluded that performing evaluations based on radiomic features is a viable option in the GAN models. 

In this study, we have used the ACRIN 6677/RTOG 0625 data set which is a multi-center dataset and is one of the strengths of this study. Of note, as a limitation, future studies with a large sample size are suggested. This study considered synthesis for two-contrast brain MRI; hence the proposed models can also be used for other related tasks in medical image analysis such as T1W, CT-PET, and MR-CT. For future research, it is suggested that relevant evaluations be carried out based on radiomic features with larger data sets and other anatomical areas. 

Despite the demonstrated effectiveness of our method in generating a T2W MRI from a FLAIR, and vice versa, it is important to acknowledge that the applicability of our approach may have limitations in certain specific cases. The performance of our method may be influenced by factors such as extreme variations in tumor size, irregular tumor shapes, or cases with substantial edema or necrosis. While our methodology has shown promising results in brain tumor patients, further research is needed to investigate its robustness in challenging scenarios and to develop additional techniques to address these limitations. Future studies should also consider expanding the dataset to include a larger cohort of patients with a wider spectrum of brain pathologies to ensure the generalizability of our findings.

## 5. Conclusions

The CycleGAN method can be used to generate realistic synthetic T2W and FLAIR brain scans, supported by both experimental qualitative and quantitative results. Radiomic features, representing quantitative data extracted from radiology images, hold a lot of promise as a novel approach to quantitatively evaluate similarities between synthetic and real MRI scans and make decisions based on the radiologic quality of the synthetic scans. Synthesis of realistic MRI scans can facilitate imaging of uncooperative patients and significantly shorten the image acquisition time thereby contributing to reducing costs to healthcare systems.

## Figures and Tables

**Figure 1 cancers-15-03565-f001:**
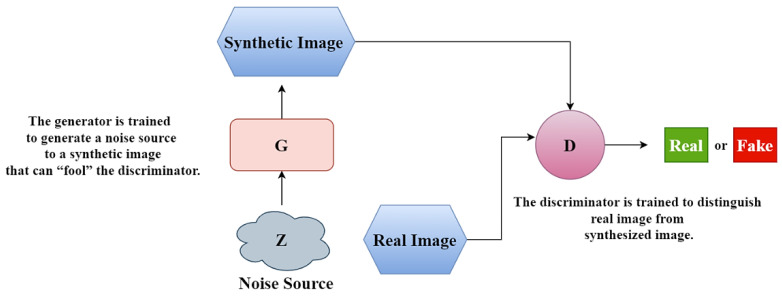
Simultaneous training of both models, the generator (G) and the discriminator (D), in competition with each other.

**Figure 2 cancers-15-03565-f002:**
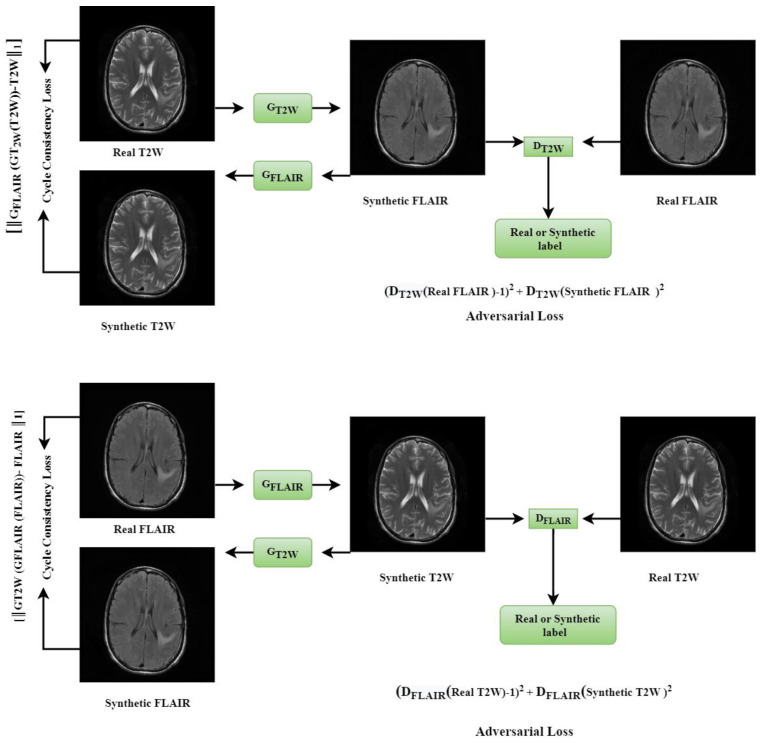
The CycleGAN model based on two mappings G_T2W_: T2W → FLAIR and G_FLAIR_: FLAIR → T2W.

**Figure 3 cancers-15-03565-f003:**
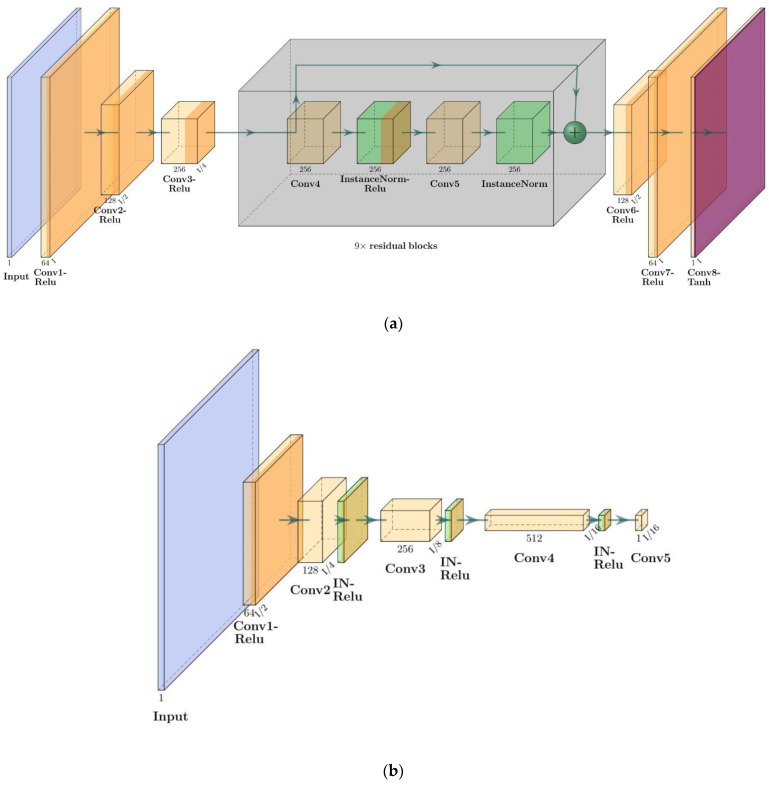
The architecture of proposed CycleGAN model; (**a**,**b**) are generator and discriminator networks respectively. Conv and IN are Convolution and Instance normalization, respectively.

**Figure 4 cancers-15-03565-f004:**
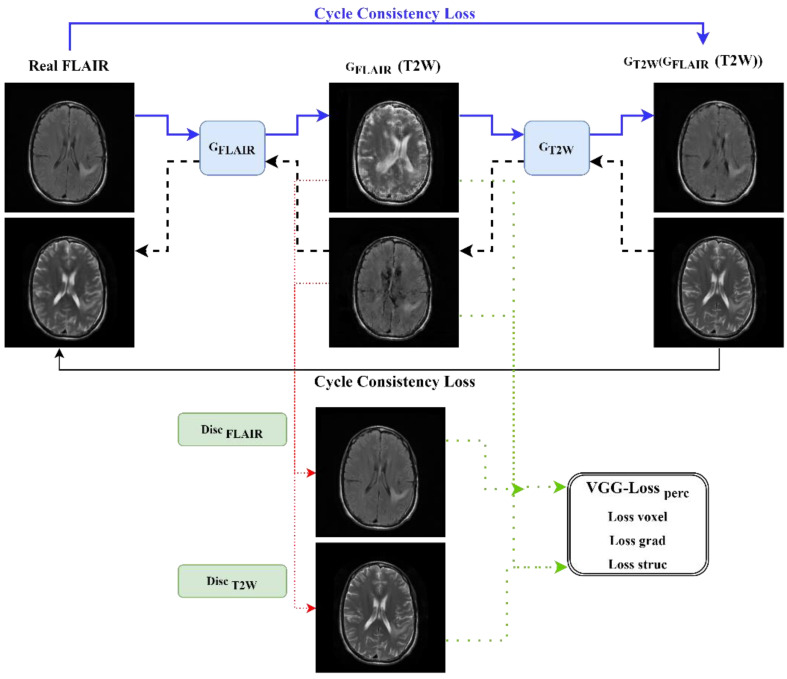
Diagram of the DC^2^Anet model.

**Figure 5 cancers-15-03565-f005:**
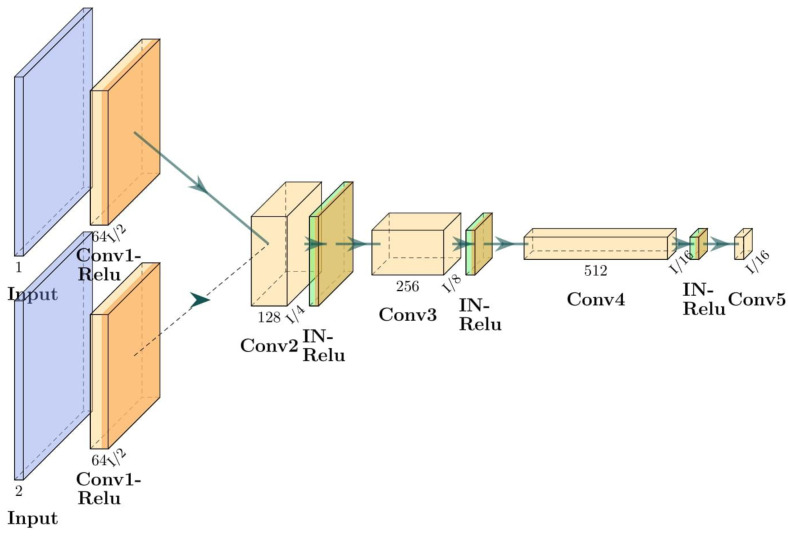
The architecture of discriminator used for DC^2^Anet model. Conv: Convolution. IN: Instance Normalization.

**Figure 6 cancers-15-03565-f006:**
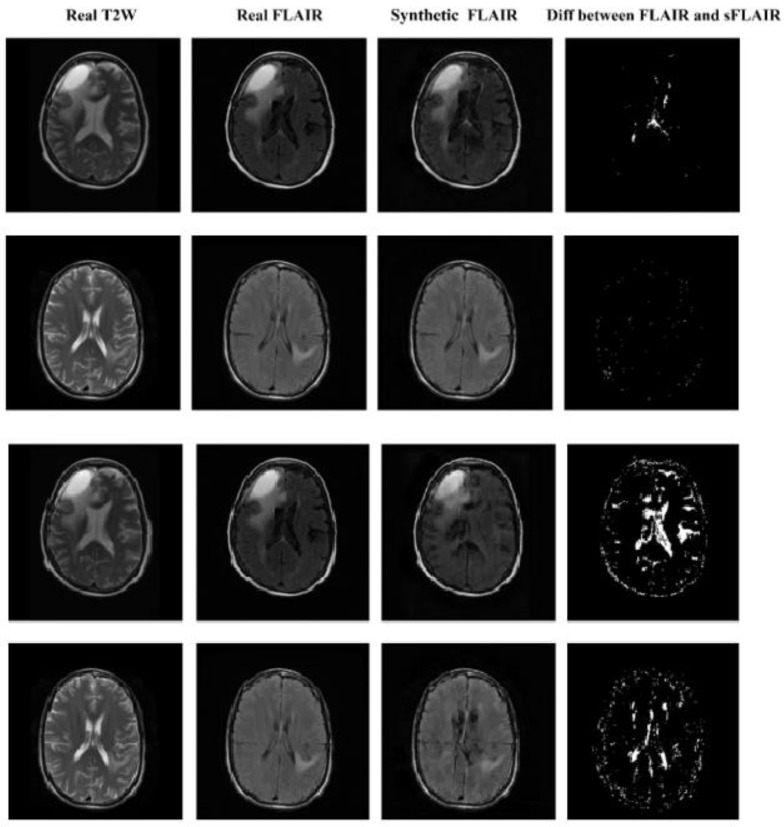
Qualitative comparison of T2W to FLAIR for CycleGAN (First and second row) and DC^2^Anet (Third and fourth row).

**Figure 7 cancers-15-03565-f007:**
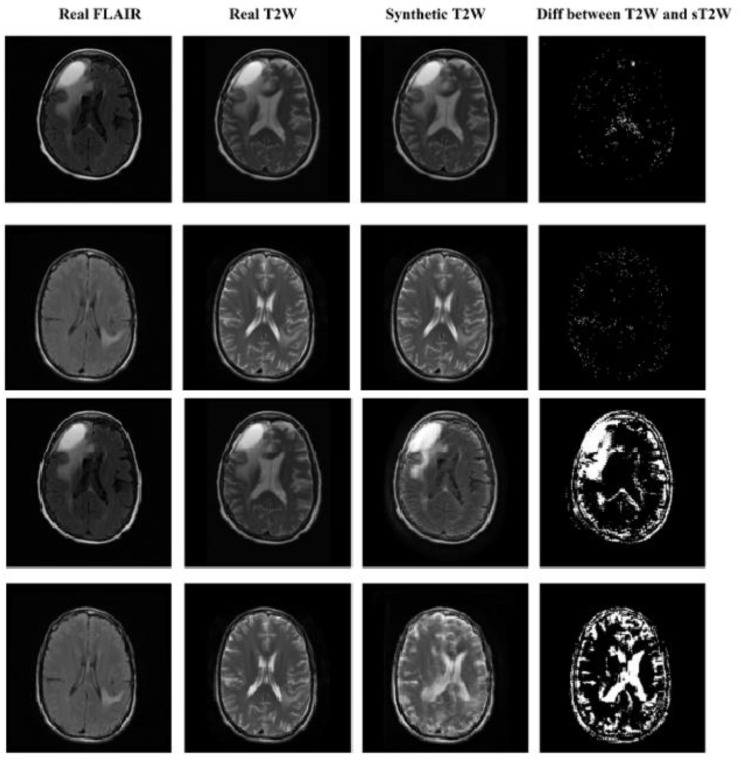
Qualitative comparison of FLAIR to T2W for CycleGAN (First and second row) and DC^2^Anet (Third and fourth row).

**Table 1 cancers-15-03565-t001:** Network Hyperparameters, Tuning, and Computing for Image Translation using CycleGAN and DC^2^Anet.

Implementation (Hyperparameter, Tuning, and Computing)	Value
Resolution (FLAIR)	256 × 256
Resolution (T2W)	512 × 512
Preprocessing	Rigid registration to ensure 256 × 256 resolution for FLAIR images
Input Image Size	256 × 256 pixels
Training Epochs	400
Batch Size	2
Optimizer	Adam
Learning Rate	2·10^−4^ (fixed for the first 200 epochs, linear decay to 0 for the next 200 epochs)
Generator Iterations	3
Discriminator Iterations	1
Cycle Consistency Loss Weight (λcyc)	10
Adversarial Loss Weight (λGAN)	1
Computing Platform	Google Colab
Programming Language	Python 3.7
Deep Learning Framework	TensorFlow 2.4.1

**Table 2 cancers-15-03565-t002:** Extracted Radiomic features for the evaluation of CycleGAN and DC^2^Anet translations.

Radiomic Features Extracted		
GLCM	GLSZM	GLRLM
Auto Correlation	Grey Level Mean	Grey Level Nonuniformity
Cluster Prominence	Grey Level Nonuniformity	High Grey Level Run Emphasis
Cluster Shade	Grey Level Variance	Long Run High Grey Level Emphasis
Contrast	High Grey Level Emphasis	Long Run Low Grey Level Emphasis
Correlation	Large Zone Emphasis	Low Gray Level Run Emphasis
Energy	Large Zone High Grey Level Emphasis	Run Length Nonuniformity
Entropy	Large Zone Low Grey Level Emphasis	Short Run Emphasis
Homogeneity	Low Grey Level Emphasis	

**Table 3 cancers-15-03565-t003:** Results of three metrics between the real T2W and synthetic T2W and real FLAIR and synthetic FLAIR, including MAE, MSE, and PSNR. Bold text highlights the best result per column.

Experiment		MAE	MSE	PSNR
Synthetic and Real T2W	CycleGAN	24.2 ± 8.1	916.6 ± 157	27.5 ± 5.6
	DC^2^Anet	35.2 ± 12	1100 ± 169	**20.5 ± 8.2**
Synthetic and Real FLAIR	CycleGAN	**17.23 ± 8.2**	**813.8 ± 198**	29.7 ± 6.55
	DC^2^Anet	25.6 ± 6.2	915.8 ± 156	21 ± 7.2

**Table 4 cancers-15-03565-t004:** GLCM-based radiomic features for real and synthetic T2W for CycleGAN and DC^2^Anet models.

Features	Real T2W	Synthetic T2W-CycleGAN	Synthetic T2W-DC^2^Anet	95% CI of Diff	
	Mean ± SE **	Mean ± SE	Mean ± SE	Real T2W vs. Synthetic T2W-CycleGAN	Real T2W vs. Synthetic T2W-DC^2^Anet
Auto Correlation	330 ± 26.40	331 ± 52.22	348 ± 49.5	−176.8 to 173.8	−194.5 to 156.1
Cluster Prominence	17,303 ± 1501	15,367 ± 1009	12,133 ± 455	−2019 to 5892	1214 to 9125 *
Cluster Shade	722.4 ± 41.14	716 ± 18.76	677.2 ± 33.74	−111.2 to 124.0	−72.39 to 162.7
Contrast	43.96 ± 0.9	42.06 ± 1.04	31.67 ± 0.86	−1.487 to 5.296	8.902 to 15.68 *
Correlation	0.005 ± 0.0004	0.0051 ± 0.0007	0.05 ± 0.0098	−0.0212 to 0.0195	−0.0703 to −0.0296 *
Energy	0.43 ± 0.01	0.44 ± 0.01	0.43 ± 0.007	−0.040 to 0.027	−0.029 to 0.037
Entropy	2.47 ± 0.08	2.4 ± 0.03	2.34 ± 0.03	−0.139 to 0.263	−0.074 to 0.328
Homogeneity	0.67 ± 0.01	0.68 ± 0.009	0.63 ± 0.007	−0.044 to 0.021	0.0052 to 0.071

* Significant difference; ** Standard error (SE).

**Table 5 cancers-15-03565-t005:** GLCM-based radiomic features for real and synthetic FLAIR for CycleGAN and DC^2^Anet models.

Features	Real FLAIR	Synthetic FLAIR-CycleGAN	Synthetic FLAIR -DC^2^Anet	95% CI of Diff	
	Mean ± SE	Mean ± SE	Mean ± SE	Real FLAIR vs. Synthetic FLAIR—CycleGAN	Real FLAIR vs. Synthetic FLAIR—DC^2^Anet
Auto Correlation	277 ± 5.69	269.6 ± 5.1	263.9 ± 5.84	−12.42 to 27.11	−6.673 to 32.86
Cluster Prominence	17372 ± 16.34	17134 ± 44.08	17036 ± 50.17	96.26 to 379.2 *	194.3 to 477.2 *
Cluster Shade	825.8 ± 42.54	819.7 ± 37.53	807.4 ± 40.61	−137.6 to 149.6	−125.2 to 162.0
Contrast	47.53 ± 1.99	45.03 ± 1.74	42.13 ± 1.78	−4.076 to 9.062	−1.166 to 11.97
Correlation	0.005 ± 0.0005	0.0005 ± 0.0003	0.05 ± 0.003	−0.0077 to 0.0076	−0.0531 to −0.0378 *
Energy	0.45 ± 0.006	0.42 ± 0.01	0.41 ± 0.01	−0.0283 to 0.0777	−0.0093 to 0.0968
Entropy	2.33 ± 0.02	2.41 ± 0.1	2.3 ± 0.11	−0.4098 to 0.2388	−0.2971 to 0.3515
Homogeneity	0.67 ± 0.01	0.65 ± 0.01	0.63 ± 0.01	−0.0287 to 0.0726	−0.0028 to 0.0985

* Significant difference.

**Table 6 cancers-15-03565-t006:** GLRLM-based radiomic features for real and synthetic T2W for CycleGAN and DC^2^Anet models.

Features	Real T2W	Synthetic T2W-CycleGAN	Synthetic T2W-DC^2^Anet	95% CI of Diff	
	Mean ± SE	Mean ± SE	Mean ± SE	Real T2W vs. Synthetic T2W-CycleGAN	Real T2W vs. Synthetic T2W-DC^2^Anet
Grey Level Nonuniformity	73.95 ± 0.58	73.89 ± 0.84	74.66 ± 0.94	−2.821 to 2.937	−3.590 to 2.168
High Grey Level Run Emphasis	60.38 ± 2.3	56.89 ± 2.43	69.63 ± 1.56	−4.241 to 11.21	−16.87 to −1.630 *
Long Run High Grey Level Emphasis	92.88 ± 1.32	92.84 ± 1.95	95.25 ± 1.94	−6.258 to 6.338	−8.670 to 3.926
Long Run Low Grey Level Emphasis	29.69 ± 0.59	28.40 ± 0.60	25.88 ± 0.29	−0.5560 to 3.138	1.973 to 5.667 *
Low Gray Level Run Emphasis	0.62 ± 0.007	0.62 ± 0.006	0.61 ± 0.004	−0.0130 to 0.0300	−0.0068 to 0.0362
Run Length Nonuniformity	53.88 ± 1.57	55.08 ± 1.18	53.63 ± 0.94	−5.690 to 3.296	−4.237 to 4.749
Short Run Emphasis	0.48 ± 0.01	0.50 ± 0.01	0.47 ± 0.01	−0.0579 to 0.0336	−0.0313 to 0.0601

* Significant difference.

**Table 7 cancers-15-03565-t007:** GLRLM-based radiomic features for real and synthetic FLAIR for CycleGAN and DC^2^Anet models.

Features	Real FLAIR	Synthetic FLAIR -CycleGAN	Synthetic FLAIR -DC^2^Anet	95% CI of Diff	
	Mean ± SE	Mean ± SE	Mean ± SE	Real FLAIR vs. Synthetic FLAIR-CycleGAN	Real FLAIR vs. Synthetic FLAIR-DC^2^Anet
Grey Level Nonuniformity	74.05 ± 0.29	71. 9 ± 0.48	74.53 ± 0.78	0.1455 to 4.144 *	−2.477 to 1.522
High Grey Level Run Emphasis	69.12 ± 1.5	68.21 ± 1.72	62.75 ± 1.23	−4.442 to 6.259	1.019 to 11.72 *
Long Run High Grey Level Emphasis	118.4 ± 5.58	115.6 ± 5.34	114.3 ± 5.29	−16.41 to 22.17	−15.10 to 23.48
Long Run Low Grey Level Emphasis	31.27 ± 0.67	29.16 ± 1	25.5 ± 0.82	−0.9038 to 5.112	2.757 to 8.773 *
Low Gray Level Run Emphasis	0.62 ± 0.006	0.61 ± 0.009	0.62 ± 0.009	−0.0205 to 0.0412	−0.0301 to 0.0317
Run Length Nonuniformity	53.23 ± 1.15	58.62 ± 3.43	58 ± 3.31	−15.50 to 4.714	−14.87 to 5.338
Short Run Emphasis	0.51 ± 0.006	0.53 ± 0.01	0.52 ± 0.01	−0.0642 to 0.0236	−0.0631 to 0.0247

* Significant difference.

**Table 8 cancers-15-03565-t008:** GLSZM-based radiomic features for real and synthetic T2W for CycleGAN and DC^2^Anet models.

Features	Real T2W	Synthetic T2W-CycleGAN	Synthetic T2W-DC^2^Anet	95% CI of Diff	
	Mean ± SE	Mean ± SE	Mean ± SE	Real T2W vs. Synthetic T2W-CycleGAN	Real T2W vs. Synthetic T2W-DC^2^Anet
Grey Level Mean	8.92 ± 0.38	9.49 ± 0.48	9.2 ± 0.48	−2.191 to 1.049	−1.905 to 1.336
Grey Level Nonuniformity	1.26 ± 0.02	1.4 ± 0.04	1.38 ± 0.03	−0.2608 to −0.0051 *	−0.2419 to 0.0137
Grey Level Variance	42.32 ± 0.57	43.40 ± 1.47	42.5 ± 1.372	−5.407 to 3.232	−4.504 to 4.134
High Grey Level Emphasis	136.1 ± 4.6	139.1 ±10.21	137.6 ± 10.27	−34.25 to 28.30	−32.82 to 29.73
Large Zone Emphasis	121.8 ± 8.08	119.9 ± 7.38	121.6 ± 7.41	−25.29 to 29.12	−27.03 to 27.38
Large Zone High Grey Level Emphasis	240.6 ± 4.04	245.6 ± 3.71	249.1 ± 3.56	−18.50 to 8.430	−22.00 to 4.930
Large Zone Low Grey Level Emphasis	152.2 ± 2.93	145.4 ± 5.44	135.1 ± 4.12	−8.474 to 22.10	1.827 to 32.40 *
Low Grey Level Emphasis	0.18 ± 0.01	0.19 ± 0.009	0.18 ± 0.008	−0.0510 to 0.0237	−0.0451 to 0.0296

* Significant difference.

**Table 9 cancers-15-03565-t009:** GLSZM-based radiomic features for real and synthetic FLAIR for CycleGAN and DC^2^Anet models.

Features	Real FLAIR	Synthetic FLAIR -CycleGAN	Synthetic FLAIR-DC^2^Anet	95% CI of Diff	
	Mean ± SE	Mean ± SE	Mean ± SE	Real FLAIR vs. Synthetic FLAIR-CycleGAN	Real FLAIR vs. Synthetic FLAIR-DC^2^Anet
Grey Level Mean	10.13 ± 0.26	10.12 ± 0.25	9.6 ± 0.19	−0.860 to 0.874	−0.349 to 1.386
Grey Level Nonuniformity	1.34 ± 0.02	1.3 ± 0.05	1.21 ± 0.06	−0.139 to 0.206	−0.047 to 0.298
Grey Level Variance	45.95 ± 2.34	47.5 ± 1.39	46.13 ± 1.24	−7.729 to 4.629	−6.357 to 6.001
High Grey Level Emphasis	152.5 ± 2.33	152.4 ± 2.66	141.3 ± 3.24	−9.836 to 9.940	1.322 to 21.10 *
Large Zone Emphasis	146.6 ± 9.91	142.8 ± 8	143.5 ± 8.1	−27.19 to 34.78	−27.88 to 34.09
Large Zone High Grey Level Emphasis	333.9 ± 8.78	331 ± 6.32	323.4 ± 7.4	−24.08 to 29.90	−16.49 to 37.49
Large Zone Low Grey Level Emphasis	147.8 ± 9.1	146 ± 8.18	119 ± 5.88	−26.05 to 29.57	0.995 to 56.61 *
Low Grey Level Emphasis	0.17 ± 0.007	0.18 ± 0.006	0.18 ± 0.005	−0.035 to 0.011	−0.032 to 0.014

* Significant difference.

## Data Availability

The data presented in this study are available in this article.

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
