# Peer review of "Within-Modality Synthesis and Novel Radiomic Evaluation of Brain MRI Scans"

_cancers, 2023, doi:10.3390/cancers15143565_

Round 1

Reviewer 1 Report

·       Please list all network hyperparameters in a sperate table

·       I cannot see comparison with similar work, how the authors can assess the performance of their method?

·       Figure 7 and figure 8 are on two sperate pages, which makes it difficult to compare them, preferably these should be fitted on a single page.

·       The authors have not cited all related work to CNN and brain tumor. For example this work is not cited [A]

[A] Abdullah, Mohammed AM, et al. "LBTSNet: A fast and accurate CNN model for brain tumour segmentation." Healthcare Technology Letters 8.2 (2021): 31-36.

Minor editing of English language required

Author Response

We would like to thank the reviewers for the constructive feedback. We have edited the manuscript to address the comments which are attached here.

Reviewer 2 Report

Within-Modality Synthesis and Novel Radiomic Evaluation of Brain MRI Scans

Seyed Masoud Rezaeijo and colleagues wrote the manuscript 'Within-Modality Synthesis and Novel Radiomic Evaluation of Brain MRI Scans'. This manuscript allows the translation of T2-weighted (T2W) magnetic resonance imaging (MRI) volumes into T2-weighted fluid-attenuated inversion recovery (FLAIR) volumes and vice versa. To evaluate the proposed method, they propose a novel evaluation scheme for generative and synthetic approaches based on radiomic features. The paper is well written. It describes well the methodology followed and the rationalization of the steps. However, in my opinion, more innovative analyses are missing from the manuscript. The main methods have been published in other types of imaging platforms. The only novel analysis is the application including the radiomic features analysis.

Major concerts

Question 1.

In the manuscript, the authors did not mention the training and validation split of the generator. Can the authors explain more this point?

Question 2.

Is the architecture of the generator and discriminator in CycleGAN a new design? Or has it been adopted from a previous publication?

Question 3.

The architecture of the generator DC 2 Anet is not mentioned. The authors said it was similar to CycleGAN. Can the authors explain it better?

Question 4.

Cycle GAN and DC 2 Anet do not include methodological details on the loss function, key components used for the generator networks, and discriminator networks for translating the T2WMRI to T2WFLAIR.

Question 5.

What are the challenges or limitations of radiomics (features) in the field of medical images in the case of their application?

Question 6.

The authors mention the evaluation of radiomics features but do not contain any information or background details about these features. They are included in Table 1, but maybe it is better to have more information as supplementary information in the manuscript.

Question 7.

Radiomic features are used to differentiate between real and synthetic images... why are these features used "8 Gray Level Co-occurrence Matrix features (GLCM)[54,55]", "8 Gray Level Size Zone (GLSZM)[56]" and “7 Gray Level Run Length Matrix feature (GLRLM)[57]” (Table 1). how these selected 23 features are relevant to the evaluation?

Question 8.

Figures 7 and 8 show different examples of T2W or FLAIR images of 5 patients each, but they don't explain the different examples very well. For example, how this method is applied in the brain with different pathologies and how it works in different patients.

Can this method maintain the size of the brain tumor for in examples of patients when the images are transformed from T2W to FLAIR or vice versa?

Question 9.

What is the origin or possible explanation for the differences between sFLAIR and real FLAIR shown in Figure 8?

Question 10.

The author found that the CycleGAN method performed better than DC2-Anet when comparing different features. However, the features are significant but were not explained well why they are different and how this affects the images

Question 11.

The paragraph does not mention any limitations or shortcomings of the proposed models or the evaluation process. Including a discussion of potential limitations, such as data variability, generalizability, or specific challenges encountered during the evaluation, would allow for a better assessment of the results.

Minor concerts

Abstract: typo error  in: “This approach allows to translate T2-Weighted (T2W) Magnetic Resonance Imaging (MRI) volume to T2-weighted-Fluid-Attenuated-Inversion-Recovery (FLAIR) volume and vica versa.’’

Figure 6. This figure is showing the segmentation and feature extraction of brain images, however, it is unclear, and it doesn’t show any good representation of what the title of the figure said. 

Table 1 and Table 2 legends could be at the bottom of the tables.

Some of the figures should be included in the supplementary data and provide more text to explain some of the points that could be improved.

Author Response

(The authors gave the same response as above.)

Round 2

Reviewer 2 Report

The authors answered all the questions and the text has been improved to explain some points better.

In the final version, the authors included two supplementary figures, but I did not find any citations in the main text about these figures. Try to have them if I'm right.